# Nivolumab and Ipilimumab Acting as Tormentors of Advanced Tumors by Unleashing Immune Cells and Associated Collateral Damage

**DOI:** 10.3390/pharmaceutics16060732

**Published:** 2024-05-29

**Authors:** Bushra Khan, Rowaid M. Qahwaji, Mashael S. Alfaifi, Mohammad Mobashir

**Affiliations:** 1Department of Biosciences, Jamia Millia Islamia, New Delhi 110025, India; bushra2008698@st.jmi.ac.in; 2Department of Medical Laboratory Sciences, Faculty of Applied Medical Sciences, King Abdulaziz University, Jeddah 22233, Saudi Arabia; rgahwajy@kau.edu.sa; 3Hematology Research Unit, King Fahd Medical Research Center, King Abdulaziz University, Jeddah 21589, Saudi Arabia; 4Department of Epidemiology, Faculty of Public Health and Health Informatics, Umm Al-Qura University, Makkah 21955, Saudi Arabia; msfaifi@uqu.edu.sa; 5Department of Microbiology, Tumor and Cell Biology (MTC), Karolinska Institute, Solnavägen 9, 171 65 Solna, Sweden

**Keywords:** ipilimumab, nivolumab, translational genomics, immune subtypes, molecular pathways, CTLA-4, PDL-1, advanced melanoma, potentially malignant tumors

## Abstract

Combining immune checkpoint inhibitors, specifically nivolumab (anti-PD-1) and ipilimumab (anti-CTLA-4), holds substantial promise in revolutionizing cancer treatment. This review explores the transformative impact of these combinations, emphasizing their potential for enhancing therapeutic outcomes across various cancers. Immune checkpoint proteins, such as PD1 and CTLA4, play a pivotal role in modulating immune responses. Blocking these checkpoints unleashes anticancer activity, and the synergy observed when combining multiple checkpoint inhibitors underscores their potential for enhanced efficacy. Nivolumab and ipilimumab harness the host’s immune system to target cancer cells, presenting a powerful approach to prevent tumor development. Despite their efficacy, immune checkpoint inhibitors are accompanied by a distinct set of adverse effects, particularly immune-related adverse effects affecting various organs. Understanding these challenges is crucial for optimizing treatment strategies and ensuring patient well-being. Ongoing clinical trials are actively exploring the combination of checkpoint inhibitory therapies, aiming to decipher their synergistic effects and efficacy against diverse cancer types. This review discusses the mechanisms, adverse effects, and various clinical trials involving nivolumab and ipilimumab across different cancers, emphasizing their transformative impact on cancer treatment.

## 1. Introduction

The development of immune checkpoint inhibitors, especially monoclonal antibodies that target vital proteins like programmed death receptor 1 (PD-1) and cytotoxic lymphocyte-associated protein 4 (CTLA-4), has revolutionized the treatment of cancer. One of these innovative medications is ipilimumab, an anti-CTLA-4 monoclonal antibody that has become a mainstay in the arsenal of treatments for cancer, including esophageal, renal, and melanoma. It is essential for suppressing CTLA-4, a coinhibitory receptor on T cells that controls the immune system in a precise manner. In doing so, ipilimumab activates T cells and unleashes a powerful immune response against cancerous cells [1]. PD1/PDL1 primarily regulates T cell activation in peripheral tissues, while CTLA4/B7 regulates T cell activation primarily in secondary lymphoid organs. By inhibiting the activation of T cells, these molecules help maintain immune homeostasis and prevent autoimmunity.

Nivolumab is a checkpoint inhibitor immunotherapy that targets PD-1 to bolster the immune system’s ability to detect and combat cancer cells (Figure 1). Approved for advanced melanoma, it enhances survival outcomes and response rates.

Whether used alone or in combination therapies, nivolumab demonstrates efficacy in adjuvant and first-line settings, offering durable responses and long-term survival benefits. Its role in melanoma treatment underscores its importance as a key therapeutic option in clinical practice [2].

Ipilimumab is a monoclonal antibody that targets CTLA-4, enhancing the immune response against cancer cells. Approved for advanced melanoma, it works by activating T cells to attack tumors. Ipilimumab has shown efficacy in improving survival outcomes and durable responses in patients with melanoma. Used alone or in combination therapies, it plays a crucial role in the treatment landscape, offering a valuable option for patients with this aggressive form of skin cancer [3].

Combining nivolumab and ipilimumab has attracted a lot of interest due to its potential for synergy, particularly when used as a first-line treatment for patients at medium and low risk [4,5,6,7,8,9,10,11,12,13,14,15]. Compared to conventional tyrosine kinase inhibitors, this combination has shown appreciable increases in both progression-free survival (PFS) and overall survival (OS), signaling a significant milestone in the field of cancer therapies. Nevertheless, despite its effectiveness, doctors struggle to manage the complexities of side effects; skin-related problems with ipilimumab therapy stand out in particular. Sweet’s syndrome is an uncommon condition that highlights the significance of careful observation and accurate handling during cancer treatment [16].

A comprehensive understanding of the immunological mechanisms behind CTLA-4 is essential to appreciating the transforming effect of immune checkpoint inhibitors. The complex ways in which CTLA-4 regulates T cell activation, proliferation, and differentiation highlight the variety of ways in which it influences immunological responses. The intricate nature of CTLA-4’s regulatory function is demonstrated by the dynamic interactions it has with different cellular constituents, including GTPases and clathrin adaptor proteins. As a therapeutic antibody, ipilimumab works against CTLA-4 in an effort to boost antitumor immunity and interfere with the complex mechanisms that cancer cells use to survive [4,5,8,9].

By blocking PD-1, the anti-PD-1 monoclonal antibody nivolumab targets a separate checkpoint concurrently. Peripheral tolerance maintenance and immune response termination are critical functions of PD-1 and its ligands, PD-L1/PD-L2. Because of its effectiveness and tolerable side effects, nivolumab is regarded as a promising cancer therapeutic agent. However, a thorough understanding is required for the best possible clinical management due to the wide range of immune-related side effects linked to these medicines, which affect several organ systems, including the central nervous system and ocular symptoms [5,6,17,18,19].

This review delves into the intricate mechanisms, potential adverse effects, and the outcomes of various clinical trials involving nivolumab and ipilimumab across different cancers.

## 2. Ipilimumab

Ipilimumab, an anti-CTLA-4 monoclonal antibody, acts on the immunological checkpoint protein CTLA-4, enhancing T cell activation and facilitating an immune response against cancer cells. This human IgG1 monoclonal antibody is utilized in the treatment of melanoma, kidney, and esophageal cancers by inhibiting CTLA4, a protein known to suppress the immune system. However, this suppression can lead to various adverse effects on the skin. In 2018, tyrosine kinase inhibitors replaced the previous first-line therapy for metastatic renal cell carcinoma [20,21].

As a coinhibitory receptor, CTLA-4 plays a crucial role in immunological homeostasis by carefully controlling T cell activity to avert autoimmunity [17,18,19,20,21,22,23,24,25]. Its dysregulation highlights its vital role in immune modulation, whether through hypofunction resulting in autoimmunity or excessive expression linked to immunosuppression in malignancy. Mainly expressed on T cells, CTLA4 coordinates early T cell activation by providing inhibitory signals and competing with CD28 for ligands, CD80/CD86, to counteract CD28 activity [17,24,25,26,27,28,29,30]. By outcompeting CD28, activating protein phosphatases like SHP2 and PP2A, and even sequestering CD80/CD86 from antigen-presenting cells, CTLA4’s increased ligand affinity limits T cell activation. Lethal immunological hyperactivation in CTLA4 knockout mice demonstrates its critical significance. CTLA4 mostly suppresses helper T cell function and increases regulatory T cell (Treg) immunosuppression, even though it is produced in CD8+ T cells. Targeted by FOXP3, CTLA4 is constitutively produced in Treg cells and when blocked, it improves immunity while preventing immunosuppression mediated by Treg cells [26]. Because of its dual effects on regulatory and effector T cells, CTLA4 is positioned as a crucial component of immunological homeostasis. Antibody therapies, such as ipilimumab, block CTLA-4 and restore antitumor immunity by blocking T cell activation and CD28:B7 interactions. The effect of ipilimumab on T cell auto-stimulation affects both side effects and therapeutic outcomes. Managing the availability of B7 ligands is a crucial aspect of the dynamic regulation of T cell activation by CD28 and CTLA4. Through its cytoplasmic domain, the co-stimulatory receptor CTLA-4 on activated T lymphocytes interacts with CD80/CD86 to transmit inhibitory signals. CD4+ T cell differentiation is influenced by CTLA-4, which coordinates T cell activation, proliferation, and differentiation [27].

By inhibiting FYN, LCK, and ZAP-70 tyrosine kinases, CTLA-4 suppresses proximal events in TCR signaling during T cell activation. The RAS pathway is constitutively activated by the hyperphosphorylation of CD35 and p52SHC in CTLA4-mutant T cells. T cell responses are improved by the antibody inhibition of CTLA-4, especially with ipilimumab, and the generation of IFN-γ is essential for anticancer immunity. Its importance is demonstrated by the presence of genetic abnormalities in IFN-γ pathway genes in non-responder patients. The degree of TCR signaling plays a complex role in regulating connections between clathrin adaptor proteins (AP-1 and AP-2) and intracellular CTLA-4 localization [31,32,33,34]. CTLA-4’s diverse regulatory function is emphasized by its association with Syp tyrosine phosphatase and its connections with the μ2 subunit of AP-2, which facilitates fast internalization. The T cell surface expression of CTLA-4 is upregulated due in part to the assembly of a CTLA-4/TRIM/LAX/Rab8 complex and the actions of different GTPases. While increased intracellular calcium levels quickly upregulate CTLA-4 on the cell surface, LRBA depletion causes CTLA-4 to degrade. To make it easier for T cells to acquire B7 ligands from APCs, CD28 mediates their trogocytosis [3,30,33,34]. By causing acquired B7 to be cis-endocytosed from the surface of T cells, CTLA4 restricts the presentation of costimulatory information. The reduction in B7 from APCs by Tregs is facilitated by a synergistic interaction between CTLA4-mediated cis-endocytosis and CD28-dependent trogocytosis. Anti-CTLA-4 antibody ipilimumab affects T cell activity, which affects therapeutic results and side effects. CTLA-4’s complex regulation mechanisms highlight the critical role it plays in immunological responses, including those related to cancer and autoimmune diseases [28]. The use of anti-CTLA-4 antibodies in particular has shown promise in boosting antitumor immunity; nevertheless, there are still issues with expanding the effectiveness of these treatments to a wider range of patient populations. The intricate relationship between CTLA-4 and T cell function as well as the tumor microenvironment emphasizes how intricate immune regulation is in cancer treatment. Together with CTLA-4 and PD-1, the B and T lymphocyte attenuator (BTLA), a glycoprotein with an immunoglobulin domain that is expressed during T cell activation, appears as a third inhibitory receptor on T lymphocytes. Tyrosine phosphorylation is induced when antigen receptors crosslink BTLAs, causing it to associate with SHP-1 and SHP-2 and reduce the generation of IL-2 [29]. Increased T cell proliferation, elevated antibody responses, and vulnerability to autoimmune encephalomyelitis are the outcomes of BTLA deficiency. A peripheral homolog of B7, B7x, ligandizes the BTLA to further define its inhibitory function in T cell responses. In a mouse model of head and neck squamous cell carcinoma (HNSCC), combining anti-CTLA4 immunotherapy with the inhibition of Src family kinases (SFKs) showed a synergistic reduction in tumor growth. The combined technique offers a promising way to increase the effectiveness of anti-CTLA4 immunotherapy in HNSCC by reducing myeloid-derived suppressor cells and Tregs while increasing the ratio of CD8+ T cells to Tregs [30,31,32,33,34,35].

In metastatic melanoma treated with CTLA-4 inhibition, TP53 mutation appears to be a potential negative predictor, linked to worse overall and progression-free survival. A possible role for FAS mRNA expression in predicting responsiveness to CTLA-4 immunotherapy is further shown by a lower expression of the gene in TP53-mutant individuals. Through cytosolic sensing of dsRNA, DNA methyltransferase inhibitors (DNMTis) induce a viral defense system in ovarian cancer, resulting in a type I interferon response and death. This reaction is accompanied by hypermethylated endogenous retrovirus (ERV) genes, and in patients with melanoma receiving immune checkpoint therapy—specifically, anti-CTLA4—the viral defense signature is correlated with a long-lasting clinical response. The findings highlight the role that DNMTis may have in influencing the immunological milieu surrounding tumors [29,31,33,35,36,37].

## 3. Nivolumab

Programmed death receptor 1 (PD1), which is found on a variety of immune cells including T cells, B cells, macrophages, and natural killer cells, is inhibited by the specialized antibody nivolumab. PD1 typically limits these cells’ ability to mount an immunological response (Figure 2). PD1 inhibits T cell function by binding to its ligands (PDL1 and PDL2) on tumor cells. This reduces T cell proliferation, cytokine generation, and cytotoxic potential [4,6,8]. Anti-PD1 antibodies were reported to occupy between 64 and 70 percent of PD1 receptors on circulating immune cells in trials assessing their pharmacodynamics. Patients with refractory metastatic cancer showed good tolerability up to the maximum planned dose of 10 mg/kg in a dose escalation study using a fully human anti-PD1 antibody (MDX 1106) [10,12,14]. Although adverse effects are seen with both anti-PD1 and anti-CTLA4 medications, preclinical models suggest that side effects associated with anti-PD1 treatment are less common and milder. The presence of B7H1 on the surface of tumor cells may function as a predictor of response to PD1 blockage, according to preliminary data from biopsies. Tumor vaccines and anti-PD1 together can also have a very synergistic effect, according to findings from animal models. Nivolumab has encouraging signs of efficacy and tolerable side effects. It works by blocking PD1, which helps the immune system target and combat cancer cells [38,39,40,41].

Immune checkpoint inhibition, which specifically targets PD-1 and its ligands PD-L1/PD-L2, has transformed the treatment of cancer and demonstrated impressive response rates for a variety of cancer types. Activated T cells produce the 288-amino-acid protein PD-1, which is essential for immune response termination that goes beyond apoptosis. The ligands for PD-1, PD-L1, and PD-L2 have different patterns of expression. Proinflammatory cytokines cause PD-L1 to be constitutively expressed on a variety of cells. Targeting the membrane-proximal extracellular region, anti-PD-1 agonist antibodies cause immunosuppressive signaling through PD-1 crosslinking. Their inhibitory effect on human T cells is enhanced by Fc engineering, indicating that these antibodies may find application in the treatment of inflammatory disorders, including autoimmune diseases. The PD-1 pathway is involved in cancer immunotherapy, organ-specific autoimmunity, and autoimmune diseases [32,33,34,35,36,37]. It is essential for self-tolerance and inflammation resolution [3]. In addition to its IgV-like domain, stalk, and cytoplasmic tail that contains an immunoreceptor tyrosine-based inhibition motif (ITIM) and switch motif (ITSM), PD-1 is a member of the CD28 superfamily and possesses unique molecular characteristics. Interleukin-2 production is suppressed and PI3K-Akt activation is inhibited by the recruitment of phosphatases SHP-1 and SHP-2 by the cytoplasmic tail tyrosines, especially Y248. The main function of PD-1 is to suppress T cell activation in the tumor microenvironment and peripheral tissues, which is essential for immunological resistance in malignancies. When T cells are activated, PD-1 induction activates its ligands (PDL1 and PDL2), which in turn inhibits T cell activation kinases through SHP2, changes the duration of T cell–APC/target cell interaction, and promotes Treg cell proliferation. Tumor-infiltrating lymphocytes (TILs) in a variety of malignancies express PD1, and its elevated expression suggests possible fatigue or anemia [38,39,40,41,42,43,44,45,46].

Tumor cell surfaces frequently express higher levels of PD1 ligands, especially PDL1, which serves as the foundation for PD1-pathway blockage, which improves antitumor effector actions. Diverse PDL1 expression patterns across various cancer types highlight the significance of comprehending the signals that trigger the production of PD1 ligands [43,44,45,46,47,48]. PDL1 on tumors is regulated by innate immune resistance and adaptive immunological resistance, which are triggered by oncogenic signaling pathways and endogenous antitumor immunity, respectively. The adaptive immune resistance theory is supported by clinical research, which highlights the significance of understanding these regulatory signals for successful PD1-pathway blocking in various cancer contexts [12,47]. A key component of tumor survival is a tumor immune escape, which is achieved by mechanisms like immunosuppression, in which the PD-L1/PD-1 signaling pathway is important. PD-L1 is regulated by the ubiquitin–proteasome system via different E3 ubiquitin ligases, such as β-TRCP, SPOP, HRD1, and STUB1. These ligases control immunological responses by coordinating the polyubiquitination and degradation of PD-L1. Reversible PD-L1 ubiquitination is catalyzed by deubiquitinating enzymes (DUBs) including CSN5, USP22, USP9X, and OTUB1. By eliminating polyubiquitin chains, CSN5 stabilizes PD-L1, and curcumin, its inhibitor, and improves CTLA-4-blocking therapy. In order to help stabilize PD-L1, USP22 directly deubiquitinates PD-L1 and modifies the CSN5-PD-L1 axis. The N-glycosylation of PD-L1 affects PD-1 interaction and protein stability. The PD-L1/PD-1 connection depends on B3GNT3-mediated poly-LacNAc, which is effectively blocked by glycosylation-specific PD-L1 antibodies, facilitating PD-L1 internalization and destruction. Strong antitumor effects are demonstrated by gPD-L1-ADC in breast cancer models. Additionally, STT3, Sigma1, and FKBP51s affect the glycosylation and stability of PD-L1, indicating possible therapeutic targets for PD-L1 expression modulation and improved anti-cancer immunotherapy. Genomic instability and the potential creation of neoantigens are caused by a mismatch repair deficit (dMMR) and microsatellite instability (MSI), both of which are correlated with positive responses to anti-PD-1/PD-L1 therapy. PD-L1 expression is influenced by oncogenic driver mutations, and the composition of the gut microbiota is becoming more widely acknowledged as a predictor of the effectiveness of anti-PD-1/PD-L1 therapy. The effectiveness of treatment is correlated with certain bacterial abundances, which may have an impact on the molecular mimicry between gut microbiota and malignancies, lymphocyte composition, dendritic cell activation, and bacterial metabolites [43,44,45,46,48,49,50,51,52,53,54]. Combining nivolumab and ipilimumab as a first-line therapy for individuals at medium and low risk was found to be efficacious in a clinical investigation. When compared to the tyrosine kinase inhibitor group, the combination showed prolonged progression-free survival (PFS) and overall survival (OS). The combination of nivolumab + ipilimumab remained superior to tyrosine kinase inhibitors in the long-term follow-up for patients with advanced renal cell carcinoma. Over the course of four years, 50% of the patients were still alive, and the hazard ratio for overall survival stayed constant. The combination of nivolumab and ipilimumab consistently had a higher response rate than the tyrosine kinase inhibitor. It is crucial to remember, nevertheless, that ipilimumab use may cause patients with melanoma to have Sweet’s syndrome. When treating patients with ipilimumab therapy, clinicians should be aware of skin lesions and fever. Corticosteroids are an essential component of the treatment of Sweet’s syndrome, albeit the precise dosage is not well established. This highlights how crucial it is to closely monitor and control any adverse effects that arise with cancer immunotherapy [2,4,5,6,7,8,9,10,17,18,19,20,21,22,23,24,25,43].

## 4. Pharmacokinetics of Ipilimumab in Combination with Nivolumab

In individuals with advanced melanoma, ipilimumab with nivolumab has been shown to be more efficacious than either medication alone. Ipilimumab is authorized as an adjuvant therapy and for advanced melanoma. The tumor response is linked to its excretion from the body, as seen with nivolumab. Anti-PD-1/PD-L1 monoclonal antibodies are associated with time-varying clearance, which is influenced by variables such as tumor size and markers of sickness severity. Anti-CTLA-4 medication such as ipilimumab also shows variable clearance over time. Reduced clearance is associated with an increase in body weight and albumin, suggesting a relationship with the severity of the illness. Concentrations of imilimumab do not change after a 12-week monotherapy period. When coupled with nivolumab, T50, or the time for concentrations to drop by half, is approximately 106 days, which corresponds with the length of the tumor response. The results of a sensitivity analysis point to additional variables affecting ipilimumab clearance. Although time-varying variables enhance the model fit, they do not fully account for the temporal impact that has been seen; this suggests that other factors are involved. Tracking the elimination of monoclonal antibodies can be used as a preemptive measure of the treatment response. Comprehending the pharmacokinetics of ipilimumab, particularly in conjunction with nivolumab, offers valuable understanding on its efficaciousness and duration of activity [37,38,43,55,56].

## 5. Clinical Trials of Combination

### 5.1. Effect on Non-Small Cell Lung Carcinoma

The treatment of non-small cell lung cancer (NSCLC) is quite difficult since it frequently has a poor prognosis when treated with traditional chemotherapy [46,48,49,50,51,52,53]. For patients without driver mutations, platinum-based treatment is the mainstay and yields mediocre response rates. Limited improvement is provided by second-line treatments involving regimens based on docetaxel or pemetrexed [54,57,58,59,60]. CheckMate 078, a phase 3 study, reaffirmed nivolumab’s sustained superiority over docetaxel in overall survival for predominantly Chinese patients with advanced NSCLC, reflecting findings consistent with global CheckMate 017/057 trials. These results underscore nivolumab’s established efficacy as a standard treatment option for previously treated NSCLC in China [60]. In patients with advanced non-small cell lung cancer (NSCLC), especially those with PD-L1 expression ≥ 1%, a pivotal phase 3 trial comparing nivolumab + ipilimumab with chemotherapy revealed a considerably longer overall survival with the immunotherapy combination. Particularly in PD-L1-negative tumors, the distinct immunological effects of CTLA4 and PD1 suppression highlight the possibility of enlisting robust anticancer immunity (Table 1). It is still difficult to find biomarkers that show that combination immunotherapy is superior to chemotherapy [60]. Nivolumab increases T cell reactivity even at low doses, indicating that it may have an effect on T cell responses in non-small cell lung cancer [60]. The CheckMate 9LA study showed that nivolumab + ipilimumab with restricted chemotherapy significantly improved overall survival for patients with advanced non-small cell lung cancer. Durable efficacy advantages and a manageable safety profile were highlighted during the trial’s 2-year follow-up, confirming its effectiveness as a first-line treatment [60]. The 5-year follow-up of the CheckMate 227 study demonstrated the superiority of nivolumab + ipilimumab in metastatic non-small cell lung cancer, with remarkable overall survival rates. Remarkably, a sizeable fraction of 5-year survivors stopped taking their medications, highlighting the possibility of a long-term clinical benefit and preserving a standard of living that is similar to that of the general public [60]. The NEOSTAR trial and CheckMate 227 subanalyses demonstrated the potential advantages of neoadjuvant ipilimumab + nivolumab in enhancing pathologic responses in operable non-small cell lung cancer [58,59,60]. In advanced non-small cell lung cancer (NSCLC), the CheckMate 227 trial confirmed the benefits of nivolumab + ipilimumab for longer overall survival, regardless of PD-L1 expression. Its remarkable long-term efficacy was further highlighted by the pooled study, which showed maintained responses after three years [58,59,60]. Cohort studies such as CheckMate 817 showed the tolerable safety and effectiveness of ipilimumab dependent on weight in combination with flat-dose nivolumab, providing advantages to a range of patient populations [58,59,60]. According to the NEOSTAR trial, surgical results following neoadjuvant treatment with nivolumab or nivolumab with ipilimumab were similar to conventional methods. In advanced non-small cell lung cancer, a network meta-analysis demonstrated the enduring advantages of nivolumab + ipilimumab in contrast to alternative regimens [53,54,57,58,59,60]. The combination’s efficacy and safety in a variety of demographics were corroborated by subanalyses from the CheckMate 9LA study, which focused on Asian patients [60]. On the other hand, the CheckMate 451 study showed that in patients with extensive-disease small cell lung cancer (SCLC), maintenance therapy consisting of nivolumab plus ipilimumab did not significantly increase overall survival when compared to a placebo. The combination demonstrated greater incidence of grade 3–4 treatment-related side events, even in the face of an increasing trend in tumor mutational load [60]. In recurrent small cell lung cancer, the CheckMate 032 trial found that although the combination exhibited a greater objective response rate, overall survival was not significantly different from that of nivolumab monotherapy [60]. CheckMate 012, which examined nivolumab + ipilimumab as first-line treatment, demonstrated encouraging clinical activity and a manageable safety profile, paving the way for more phase 3 investigations [58,59,60]. The results of the STIMULI study highlight the need for caution when interpreting consolidation immunotherapy results following chemo-radiotherapy in patients with small-cell lung cancer with limited disease. The median time-to-treatment-discontinuation draws attention to toxicity and cessation-related issues that require careful thought [58,59,60].

### 5.2. Effect on Esophageal Carcinoma

Metastatic esophagogastric cancer presents a significant global health challenge with limited treatment options post-second-line chemotherapy. Nivolumab, a programmed death-1 (PD-1) inhibitor, has shown promising results in both Asian and Western populations. In Asian patients with advanced stomach or gastroesophageal junction tumors, nivolumab demonstrated improved overall survival compared to a placebo [63]. Similarly, in the Western population with chemotherapy-refractory gastroesophageal malignancy, nivolumab alone and in combination with ipilimumab exhibited clinically meaningful antitumor activity (Table 2) [61,63,64,65,66].

Nivolumab’s efficacy extends beyond monotherapy, as demonstrated in the CheckMate-032 trial. This study evaluated nivolumab alone and in combination with ipilimumab in Western patients with metastatic esophagogastric cancer refractory to chemotherapy. The trial showed durable responses and encouraging long-term overall survival, supporting a further evaluation of these immunotherapies in earlier lines of therapy [66,67,68]. Furthermore, in the CheckMate 648 trial, first-line treatment with nivolumab plus chemotherapy and nivolumab plus ipilimumab significantly improved overall survival in previously untreated advanced esophageal squamous-cell carcinoma compared to chemotherapy alone [66,67,68].

The efficacy of nivolumab and ipilimumab combination therapy extends to different patient populations. In a Japanese subgroup analysis of the CheckMate 648 trial, first-line treatment with nivolumab plus ipilimumab or nivolumab plus chemotherapy demonstrated significant survival benefits compared to chemotherapy alone in patients with advanced esophageal squamous-cell carcinoma [65,66,67,68]. Additionally, in the Chinese subgroup analysis of the phase 3 CheckMate 649 study, first-line treatment with nivolumab plus chemotherapy exhibited significant efficacy improvements compared to chemotherapy alone in patients with advanced gastric, gastroesophageal junction, and esophageal adenocarcinoma [62,63,64,65,66,67,68].

Despite the promising potential of nivolumab plus ipilimumab, the combination did not meet significance in overall survival improvement in the CheckMate 649 trial, highlighting the need for further investigation into its efficacy and safety profile. However, the RAMONA trial, a multicenter phase 2 study, demonstrated the potential benefits of combined nivolumab and ipilimumab as second-line therapy in elderly patients (≥65 years) with advanced esophageal squamous-cell carcinoma. The combination therapy showed a median overall survival of 7.2 months, significantly higher than historical cohorts receiving standard chemotherapy [65,66,67,68].

**Table 2 pharmaceutics-16-00732-t002:** Summary of completed clinical trials of ipilimumab plus nivolumab in esophageal carcinoma [62,66,69].

Reference	Trial Phase	Treatment Arms	Primary Endpoints	Results	Clinical Trial Number
Kato et al., 2022	Phase 3	Nivolumab + Ipilimumab (NIVO + IPI), Nivolumab + Chemotherapy (NIVO + Chemo), Chemotherapy (Chemo)	Overall survival (OS), Progression-free survival (PFS)	First-line NIVO + IPI and NIVO + Chemo treatments demonstrated substantial survival advantages over Chemo in Japanese patients with advanced ESCC. Both NIVO + IPI and NIVO + Chemo arms showed acceptable tolerability.	NCT03143153
Meindl-Beinker et al., 2019	Phase 2	Nivolumab and Ipilimumab vs. Nivolumab Alone	Overall survival (OS), Time to QoL deterioration, Tumor response, Progression-free survival (PFS), Safety	The RAMONA trial demonstrated a significant survival benefit of nivolumab/ipilimumab in advanced ESCC compared to historical data of standard chemotherapy. Nivolumab/ipilimumab combination therapy showed promising results with acceptable safety profile in elderly patients with ESCC.	NCT03416244
Janjigian, Y. Y., et al. (2018)	Phase 1/2	Nivolumab, Nivolumab + Ipilimumab	Objective response rate in patients with chemotherapy-refractory esophagogastric cancer	Nivolumab and nivolumab plus ipilimumab showed clinically meaningful antitumor activity, durable responses, encouraging long-term OS, and a manageable safety profile.	NCT01928394
Shitara, K. et al. (2022)	Phase 3	Nivolumab + Chemotherapy, Nivolumab + Ipilimumab, Chemotherapy	Overall survival in patients with gastroesophageal cancer	Nivolumab + chemotherapy showed improvement in overall survival compared to chemotherapy alone. No significant improvement in overall survival was observed with nivolumab + ipilimumab.	NCT02872116

### 5.3. Effect on Renal Cell Carcinoma

In recent years, a transformative shift in the treatment landscape of metastatic renal cell carcinoma (mRCC) has emerged through the inhibition of vascular endothelial growth factor (VEGF) and immune checkpoint inhibitors (ICIs), specifically nivolumab and ipilimumab. Sunitinib, an early tyrosine kinase inhibitor targeting VEGF, established itself as a robust standard of care following its superiority over interferon alfa in 2007 [7,8,9,10,11,12,13,14,15,16,17,18,19,20,21,22,23,24,25,26,27,28,29,30,31,32,33,34,35,36,37,38,39,40,41,42,43,44,45,46,47,48,49,50,51,52,53,54,55,56,57,58,59,60,62,63,64,65,66,67,68,69,70,71,72,73,74,75,76,77]. Subsequently, the combination of ipilimumab and nivolumab gained regulatory approval, particularly recommended for medium-to-low-risk patients. The pivotal CA209214 trial led to the approval extension of nivolumab in conjunction with ipilimumab for first-line treatment in advanced/intermediate-risk renal cell carcinoma in 2018 (Table 3) [70,71,75,76,77,78,79]. This decision was rooted in the study’s phase 3 design, comparing nivolumab plus ipilimumab against sunitinib in patients with previously untreated severe renal cell carcinoma. However, the diverse therapeutic techniques arising from these advancements present challenges in categorizing the disease response, emphasizing the need for continued research. Limited evidence exists regarding the correlation between radiographic tumor recurrence and clinical judgment. Therefore, physicians’ assessments and radiological evaluations remain crucial for adapting effective therapy protocols [76,77]. The COSMIC-313 trial explored the efficacy and safety of cabozantinib in combination with nivolumab and ipilimumab compared to nivolumab and ipilimumab alone in advanced clear-cell renal cell carcinoma. The experimental group demonstrated significantly longer progression-free survival (PFS) and a higher response rate. Despite increased adverse events, the study suggests improved PFS with the combination therapy [75,76,77,78,79,80]. The CheckMate 214 trial, with over 5 years of follow-up, affirms the enduring benefits of nivolumab plus ipilimumab over sunitinib in first-line treatment for advanced renal cell carcinoma. The combination maintained superior overall survival and progression-free survival across diverse risk categories, establishing it as a long-term, first-line treatment option [74,75,76,77]. In the phase 3 CheckMate 914 trial, adjuvant therapy with nivolumab plus ipilimumab was compared to a placebo in localized renal cell carcinoma at high risk of recurrence after nephrectomy. With a median follow-up of 37 months, adjuvant nivolumab plus ipilimumab did not demonstrate improved disease-free survival compared to the placebo. Adverse events were higher, indicating caution in recommending this regimen for adjuvant treatment [76,77,78,79,81,82]. The BIONIKK trial investigated first-line treatments for metastatic clear-cell renal cell carcinoma, comparing nivolumab, nivolumab–ipilimumab, and VEGFR tyrosine kinase inhibitors. Results showed varied objective response rates across molecular subgroups, emphasizing the potential of molecular profiling for personalized treatment selection. However, adverse events, including serious ones, were observed, highlighting the importance of balancing efficacy and safety [76,77]. The CheckMate 920 trial demonstrated encouraging antitumor activity with nivolumab plus ipilimumab in advanced non-clear-cell renal cell carcinoma (nccRCC), a population often excluded from previous trials. In the FRACTION-RCC trial, the combination showed a potential salvage strategy after progression on immuno-oncology therapies, providing insights for optimal treatment sequencing [75,76,77,78,79,82]. The CheckMate 214 trial’s 42-month follow-up reaffirms the durable efficacy benefits of first-line nivolumab plus ipilimumab over sunitinib in advanced renal cell carcinoma. The combination maintains its favorable risk–benefit profile, supporting its status as a preferred first-line treatment. In the CheckMate 920 trial for metastatic non-clear-cell renal cell carcinoma, nivolumab plus ipilimumab demonstrated encouraging antitumor activity with manageable safety, suggesting potential efficacy in this subtype. The TITAN-RCC trial emphasizes personalized immunotherapeutic approaches in metastatic renal cell carcinoma, showing that the combination of nivolumab plus ipilimumab significantly enhances the objective response rate compared to nivolumab monotherapy. The discontinued arm of the CheckMate 9ER trial evaluated the nivolumab plus ipilimumab plus cabozantinib triplet regimen, showing clinical activity with overlapping toxicities, highlighting the need for careful monitoring in future studies [76,77,78,79,82,83]. In the CheckMate 214 trial’s exploratory analysis, first-line nivolumab plus ipilimumab demonstrated favorable outcomes in patients with advanced renal cell carcinoma without prior nephrectomy, suggesting potential benefits in this specific population [79,82]. A cost-effectiveness analysis comparing first-line treatments for mRCC suggests that nivolumab plus ipilimumab is cost-effective for intermediate- and poor-risk patients. Subgroup analyses highlight its pronounced cost-effectiveness in patients with a programmed cell death 1 ligand 1 expression of at least 1% [79,82]. In a randomized controlled trial comparing nivolumab plus ipilimumab with sunitinib in patients with advanced renal cell carcinoma, durable clinical benefits and higher conditional survival rates were observed with nivolumab plus ipilimumab, highlighting its long-term efficacy and potential as a first-line immunotherapy-based combination. In a randomized controlled trial, nivolumab plus ipilimumab exhibited prolonged overall survival and durable responses compared to sunitinib in patients with advanced sarcomatoid renal cell carcinoma, highlighting its efficacy as a preferred first-line therapy [75,76,77,78,79,81,82,83,84].

### 5.4. Effect on Melanoma Cancer

The combination of immune checkpoint inhibitors targeting cytotoxic T lymphocyte-associated antigen 4 (CTLA-4) and programmed death 1 (PD-1) has demonstrated heightened efficacy in advanced melanoma, albeit with notable side effects. Comparative studies of ipilimumab plus nivolumab, nivolumab alone, and ipilimumab alone reveal superior outcomes for the combination and nivolumab alone, emphasizing improved response rates, progression-free survival, and overall survival [85,86,87,88,89]. Sequential immunotherapy followed by targeted therapy proves to be beneficial in untreated BRAFV600-mutant metastatic melanoma, highlighting the potential for tailored treatment sequences. The 6.5-year follow-up of the CheckMate 067 trial underscores sustained clinical advantages for nivolumab plus ipilimumab and nivolumab alone over ipilimumab alone, revealing a remarkable median overall survival of 72.1 months. Adjuvant therapy with nivolumab exhibits prolonged recurrence-free survival and fewer adverse events compared to ipilimumab in resected stage IIIB, IIIC, or IV melanoma. Neoadjuvant ipilimumab plus nivolumab in macroscopic stage III melanoma yields high survival rates, particularly in pathologic responders [87,88,89,90,91]. The CheckMate 915 trial (Table 4) indicates nivolumab alone as a standard adjuvant treatment, and the SECOMBIT trial suggests the clinical benefit of sequential immunotherapy and targeted therapy in BRAFV600-mutant melanoma. The evolving landscape of immunotherapeutic strategies in advanced melanoma is highlighted by FDA-approved agents, nivolumab and pembrolizumab, along with ipilimumab, contributing to improved median overall survival in metastatic melanoma. Challenges of resistance mechanisms persist, driving ongoing research endeavors to enhance therapeutic approaches, addressing the diversity of resistance mechanisms to optimize patient outcomes [87,88,89,90,91]. The PRADO trial investigates personalized treatment strategies following neoadjuvant ipilimumab and nivolumab in high-risk stage III melanoma. Achieving a major pathologic response (MPR, ≤10% viable tumor) in the index lymph node allowed the omission of therapeutic lymph node dissection and adjuvant therapy, resulting in improved relapse-free survival and reduced surgical morbidity. These findings support the feasibility and benefits of response-directed treatment personalization after neoadjuvant ipilimumab and nivolumab, offering a potential paradigm for tailored therapeutic approaches in clinical practice [91,92,93,94]. In a randomized trial comparing neoadjuvant versus adjuvant ipilimumab plus nivolumab in macroscopic stage III melanoma, neoadjuvant therapy demonstrated feasibility and induced pathological responses in 78% of patients, with no relapses during a median follow-up of 25.6 months. However, both neoadjuvant and adjuvant arms experienced high-grade adverse events, emphasizing the need for further investigation to optimize efficacy while minimizing toxicity in the use of ipilimumab and nivolumab for melanoma treatment. The study suggests potential benefits of neoadjuvant application but underscores the importance of refining the regimen for improved safety [91,92,93,94]. In a phase 1 dose-escalation trial, the combination of nivolumab and ipilimumab demonstrated a remarkable 61% objective response rate in patients with advanced melanoma harboring BRAF wild-type tumors, compared to 11% with ipilimumab alone. The combination also exhibited superior median progression-free survival (not reached vs. 4.4 months) and an acceptable safety profile, emphasizing its efficacy in treatment-naive patients. This study supports the use of nivolumab plus ipilimumab as a potent first-line therapeutic option for advanced melanoma [92,93,94]. In a randomized phase 2 trial for metastatic melanoma refractory to a PD-1 blockade, the combination of ipilimumab and nivolumab demonstrated a significant improvement in progression-free survival compared to ipilimumab alone (HR = 0.63). The objective response rate also favored the combination (28% vs. 9%), highlighting the potential of reversing primary resistance to a PD-1 blockade through dual CTLA-4 and PD-1 inhibition in some patients. The study supports the exploration of combination strategies in overcoming resistance to immune checkpoint therapies. In a phase 2 study, combination nivolumab and ipilimumab, along with nivolumab monotherapy, demonstrated efficacy in treating melanoma brain metastases. The combination achieved a high intracranial response rate (46%), including complete responses, making it a promising first-line therapy for patients with asymptomatic untreated brain metastases [88,90,91,92]. In a phase 2 study, the combination of nivolumab and ipilimumab showed promising activity in metastatic uveal melanoma, with an overall response rate of 18%, including complete and partial responses. The regimen demonstrated deep and sustained confirmed responses, providing a potential treatment option for this challenging malignancy. The 3-year follow-up data from the CheckMate 204 trial evaluating nivolumab plus ipilimumab in melanoma brain metastases showed a durable response with notable clinical benefits in asymptomatic patients. The study supports the first-line use of this combination in asymptomatic individuals, highlighting its potential in improving overall survival and progression-free survival. Symptomatic patients exhibited challenges in treatment, but some achieved long-term responses, emphasizing the complexity of managing melanoma brain metastases [85,86,87,88,90,91,92,93,94,95,96,97].

### 5.5. Effect on Other Cancers

In recent studies, the combination therapy of ipilimumab and nivolumab has demonstrated efficacy in various cancers, including melanoma, non-small cell lung cancer, renal cell carcinoma, urothelial carcinoma, Hodgkin lymphoma, and microsatellite-instability-high/dMMR metastatic colorectal cancer (MSI-H/dMMR mCRC). For instance, the CheckMate 142 trial exhibited remarkable objective response rates and disease control rates in patients with MSI-H/dMMR mCRC, indicating the potential of this combination regimen as a well-tolerated treatment option. Moreover, the SWOG S1609 DART trial highlighted the efficacy of ipilimumab plus nivolumab in various rare tumors, particularly in high-grade neuroendocrine carcinomas. In advanced hepatocellular carcinoma (HCC), the combination therapy of nivolumab plus ipilimumab has shown promising results, leading to accelerated approval in the US and improved overall survival rates. Additionally, an exploratory analysis revealed that increased ipilimumab exposure enhances efficacy outcomes, supporting its use as a second-line treatment in advanced HCC [96,97,98,99].

Furthermore, in stage III urothelial cancer, preoperative immunotherapy with anti-PD-1 and anti-CTLA-4 antibodies demonstrated feasibility and efficacy, with high rates of surgical resection and the pathological complete response. Additionally, the phase 1B NABUCCO trial demonstrated the superiority of high-dose ipilimumab plus nivolumab in achieving pathological complete response rates, suggesting a potential prognostic marker for progression-free survival. Moreover, the TITAN-TCC phase 2 trial investigated nivolumab induction followed by high-dose ipilimumab as an immunotherapeutic boost in metastatic urothelial carcinoma, showing meaningful clinical activity and supporting dual checkpoint inhibition in first-line treatment. However, further investigations are warranted to optimize treatment strategies, particularly in the second-line setting [95,98,99,100,101,102,103,104,105,106].

In advanced, chemotherapy-refractory metaplastic breast cancer (MpBC), the ipilimumab and nivolumab combination demonstrated promising clinical activity with durable responses, suggesting further investigation in MpBC. Additionally, the ICON trial evaluated the combination of ipilimumab and nivolumab with anthracycline-based chemotherapy in metastatic hormone-receptor-positive breast cancer (HR + mBC), showing meaningful responses after chemotherapy. The phase 1/2 trial assessed the safety and efficacy of combining brentuximab vedotin with nivolumab or ipilimumab, or both, in patients (Table 5) with relapsed or refractory Hodgkin lymphoma, showing promising overall response rates and manageable toxicity profiles across treatment arms [106].

The phase 1b trial assessed the safety and efficacy of combining a PD-1 blockade with a CTLA-4 or KIR blockade in patients with relapsed/refractory lymphoid malignancies, indicating manageable toxicity profiles but no significant improvement in efficacy compared to a single-agent PD-1 blockade across the studied diseases [106].

Overall, these studies underscore the potential of an immune checkpoint blockade in various cancers and highlight the importance of further investigations to optimize treatment strategies and improve therapeutic outcomes.

### 5.6. Resistance to Immune Checkpoint Therapy

Resistance to nivolumab and ipilimumab, immune checkpoint inhibitors, poses a challenge in cancer therapy. Mechanisms of resistance include alterations in the tumor microenvironment, upregulation of alternative immune checkpoints, loss of tumor antigen expression, and activation of alternative signaling pathways [107,108,109]. Combination therapies, incorporating targeted agents or other immunotherapies, show promise in overcoming resistance and enhancing treatment efficacy [107,108,109,110]. Research on predictive biomarkers for response and resistance to nivolumab and ipilimumab aims to personalize treatment strategies and improve patient outcomes. Understanding these resistance mechanisms is crucial for developing novel approaches to combat treatment resistance and improve the effectiveness of immune checkpoint inhibitors in cancer therapy. Future studies should further elucidate the complex interplay of factors contributing to resistance and develop tailored therapeutic strategies to overcome it [110,111,112,113,114].

### 5.7. Effect of Immunotherapy on Immunosuppressive MDSCs

Myeloid-derived suppressor cells (MDSCs) play a pivotal role in tumor immune evasion by suppressing antitumor immune responses. Studies suggest that MDSC levels could impact the response to immunotherapy, including checkpoint inhibitors like nivolumab and ipilimumab, with elevated MDSC levels correlating with resistance to these therapies. While specific clinical data on targeting MDSCs with nivolumab and ipilimumab are limited, ongoing trials investigate the potential of KIT inhibitors in combination with immune checkpoint inhibitors to enhance antitumor immune responses by modulating MDSC activity [112,113,114,115].

All-trans retinoic acid (ATRA) has shown promise in inhibiting MDSC expansion and function, thereby enhancing the antitumor immune response. Combining ATRA with checkpoint inhibitors like nivolumab and ipilimumab may synergistically counter immune suppression mechanisms in the tumor microenvironment, potentially leading to improved treatment outcomes [113,114,115,116]. In patients with metastatic melanoma refractory to ipilimumab, nivolumab exhibited a response rate of 30%, with durable responses and favorable overall survival rates. Notably, patients with prior severe toxicity to ipilimumab tolerated nivolumab well, suggesting its potential as a viable treatment option in this population. In melanoma therapy, significant changes in MDSC levels have been observed, particularly in patients with progressive disease. The inhibition of Notch1 has shown promise in reducing MDSCs and regulatory T cells, potentially enhancing CD8+ T cell activity and improving responses to immune checkpoint inhibitors [114,115,116,117]. The interaction between MDSCs and immune checkpoint inhibitors in non-small cell lung cancer (NSCLC) is also crucial. Understanding the dynamics of MDSC subsets in patients with NSCLC receiving combination immunotherapy may provide insights into treatment resistance mechanisms and inform the development of novel therapeutic approaches [116,117,118,119]. Moreover, in regionally advanced melanoma, baseline levels of MDSCs and regulatory T cells have prognostic value, with lower levels correlating with improved relapse-free survival. A CTLA4 blockade led to transient changes in Treg and MDSC levels, suggesting sustained therapeutic effects [115,116,117,118,119].

## 6. Adverse Effects

Nivolumab and ipilimumab, among other immunotherapy agents, can induce immune-related adverse events (irAEs) ranging from mild to severe, including fatigue, rash, diarrhea, and pneumonitis. Clonal Epstein–Barr virus (EBV) and pre-existing cancer immunity may contribute to neurological complications, with male patients at higher risk. Neuromuscular disorders like myasthenia gravis or myositis can occur, challenging diagnoses and management [3]. Gastrointestinal irAEs, particularly with ipilimumab, may coincide with ocular complications such as uveitis and orbital myositis. Although rare, necrotizing severe myositis may develop, necessitating specialized treatment like immunoglobulins or plasmapheresis [119,120,121,122,123,124]. Despite these risks, the long-term clinical benefits of immunotherapy often outweigh adverse effects, emphasizing the need for vigilant monitoring and timely intervention to balance antitumor immunity with autoimmune complications. In clinical trials, the combination of nivolumab and ipilimumab in previously untreated advanced melanoma resulted in a higher incidence of treatment-related adverse events of grade 3 or 4 compared to monotherapy, particularly hepatic toxic events, with rates of grade 3 or 4 elevations of the aminotransferase levels being notably higher in the combination group (Figure 3 and Table 6 and Table 7) [114,117,118].

In the long-term follow-up of patients with advanced melanoma, treatment-related adverse events, including grade 3 or 4 events, were observed in 59% of those receiving nivolumab plus ipilimumab, 23% receiving nivolumab alone, and 28% receiving ipilimumab alone. The median time to resolution of select adverse events was generally less than 12 weeks, with exceptions noted for specific categories, highlighting the need for vigilant monitoring and management of adverse effects during treatment with these agents. Sweet’s syndrome, a rare dermatologic complication, has been documented in patients receiving ipilimumab and nivolumab combination therapy for various cancers. This adverse event underscores the importance of vigilance and prompt management of dermatologic manifestations in individuals undergoing immune checkpoint inhibitor treatment [120,121].

In the perioperative setting for resectable hepatocellular carcinoma (HCC), both nivolumab alone and nivolumab plus ipilimumab demonstrated manageable safety profiles, with 77% and 86% of patients experiencing treatment-related adverse events (trAEs), respectively. Grade 3 trAEs were higher with nivolumab plus ipilimumab (43%) compared to nivolumab alone (23%). Common trAEs included elevated liver enzymes, anemia, fatigue, and gastrointestinal symptoms. No surgical cancellations were due to trAEs, indicating feasibility of perioperative immunotherapy in HCC [23,71,99,122]. In a randomized phase 3 trial comparing nivolumab plus ipilimumab versus nivolumab alone for advanced squamous cell lung cancer, no evidence of a QOL benefit was found with the addition of ipilimumab. Patients with high levels of appetite loss and shortness of breath experienced a threefold increased risk of progression or death, while those with both appetite loss and work limitations had a fivefold increased risk of death [122].

In a phase 3 trial for advanced squamous non-small cell lung cancer, the addition of ipilimumab to nivolumab did not significantly improve overall survival compared to nivolumab alone. Grade 3 or higher treatment-related adverse events occurred in 39.5% of patients with nivolumab plus ipilimumab and 33.3% with nivolumab alone, with immune-related adverse events being common in both groups [48,66,123,124,125,126].

In patients with advanced melanoma previously progressing on a PD-1 blockade, both ipilimumab alone and ipilimumab plus nivolumab showed objective responses, with comparable disease control rates (DCRs) and rates of grade 3–4 adverse events (56% vs. 50%, respectively). Biomarker analyses indicated increased CD4+ T cells with higher polyfunctionality in patients with a clinical benefit [124].

Several studies have highlighted the significance of immune phenotypes as predictors of immune-related adverse events (irAEs) in patients (Table 6 and Table 7) with melanoma undergoing ipilimumab and nivolumab therapy. Notably, elevated levels of activated CD4+ and CD8+ T cells, along with increased proportions of effector memory T cells, have been associated with the occurrence of irAEs, suggesting a potential link between the treatment response and adverse events [117,118,124].

## 7. Future Perspectives and Conclusions

Cancers often use various strategies to evade detection and destruction by the immune system. They can manipulate antigen presentation processes, including the upregulation of MHC class I molecules or the suppression of antigen-processing mechanisms. Additionally, tumors can disrupt signal transduction pathways that control T cell inhibition and activation, recruit immunosuppressive cell types like regulatory T cells (Tregs) and myeloid-derived suppressor cells (MDSCs), and distort immune checkpoints. Conventional cytokine therapies, such as Interleukin-2 (IL-2) and interferon (IFN), have been employed for cancer treatment, but their lack of target specificity and associated side effects limit their effectiveness. Alternative forms of cancer treatments are being explored, including T cell activation pathway agonists, T cell checkpoint inhibitors, and cytokines like IL-15 and IL-12. Therapeutic vaccines, immune cell modulators, and other approaches to enhance immune cell activity are also being investigated to improve overall efficacy and patient survival. Combining immunotherapies that target different immune pathways has proven to be a promising approach. Immune checkpoint inhibitors like PD-1/L1 and CTLA-4 have revolutionized cancer treatment, but their effectiveness varies among patients. Combinations like the dual PD-1/CTLA-4 antagonist (ipilimumab/nivolumab) have shown enhanced response rates, but severe toxicity has limited their widespread use. Various combinations involving different immunotherapies have been explored, but not all have been successful. For instance, combining ipilimumab with IL-2 or Peg-IFN did not show improved efficacy compared to ipilimumab alone. Tremelimumab, another CTLA-4-targeting monoclonal antibody, failed to outperform conventional treatment in a phase 3 clinical trial. The role of innate immunity in supporting adaptive immune responses is being investigated. Natural killer (NK) cells, part of innate immunity, play a crucial role. Tumor cells can evade NK cell surveillance by imitating normal cells, which express inhibitory receptors. Cytokines like IL-21 have been studied for their role in activating NK and T cells. Combining IL-21 with immune checkpoint inhibitors has shown increased antitumor efficacy in preclinical models. Indoleamine-pyrole-2,3-dioxygenase (IDO) is an immunosuppressive enzyme that hinders both innate and adaptive immune cells. Inhibiting IDO has shown promising results in preclinical studies, and combining IDO inhibitors with other immunotherapies is being explored in clinical trials. Vaccines, designed to prepare the immune system against tumor antigens, are potential partners for combination therapies. Some combinations, like ipilimumab with granulocyte–macrophage colony-stimulating factor (GM-CSF) and other vaccines, have shown encouraging results in clinical trials. Adoptive cell transfer (ACT), involving the collection and expansion of tumor-infiltrating lymphocytes (TILs) from patients, has demonstrated long-term responses in melanoma. Other techniques, such as CAR therapy and TCR gene therapy, involve modifying patients’ T cells to induce immune reactivity against tumors. Combining these advanced techniques with other immunotherapies may further enhance clinical effectiveness. Targeted medicines can make tumor cells more susceptible to immune-mediated death by modulating various signaling pathways. Some drugs may boost efficient dendritic cell maturation, T cell priming, and the formation of long-lived memory T cells. BRAF inhibitors, which strengthen the immune system, and MEK inhibitors, which may suppress T lymphocyte and dendritic cell activity, have different effects. Understanding the impact of these drugs on signaling pathways like mTOR, PIP3/akt, PTEN phosphatases, P16, and P15, as well as their effects on apoptosis and necrotic pathways, is crucial. Telomerase, an enzyme conferring replicative potential to tumor cells, can be targeted to investigate its role in inhibiting tumor growth. Epithelial-to-mesenchymal transition (EMT) is a key driver of metastasis, and studying transcription factors involved in EMT can provide insights into the drug targets. In summary, the evolving landscape of cancer immunotherapy involves a multitude of approaches, and combining different strategies holds promise for improving patient outcomes. Investigating the intricate interactions within the immune system and comprehensively understanding the impact of these therapies on various molecular pathways are crucial steps towards developing more effective and targeted cancer treatments.

Immunotherapy is a promising approach to treating various cancers, offering the potential for long-term survival. Researchers are actively exploring ways to optimize this science by studying the combination of different immunotherapies that target distinct immune networks, as well as their combination with existing treatment methods. Understanding the appropriate timing for these combination treatments is crucial. However, identifying the best strategies is challenging due to limited information and unexpected toxicities in some combinations. Preliminary clinical data will guide ongoing research efforts. Combining therapeutic approaches has the advantage of overcoming the mechanisms that cancer cells use to evade the immune system, potentially leading to improved overall survival for a broader group of patients. Determining the optimal sequence, timing, and combination of immunotherapy is essential. It is also crucial to select the right dose and sequence. Since radiation, chemotherapy, and targeted drugs have unique mechanisms of action, their order in combination with immunotherapy requires careful consideration. Finding regimens with the best balance between risks and benefits is a final consideration when combining immunotherapies. As more drugs targeting different aspects of tumor-associated immunosuppression are discovered and used together, we anticipate advancements in overall clinical effectiveness. 

## Figures and Tables

**Figure 1 pharmaceutics-16-00732-f001:**
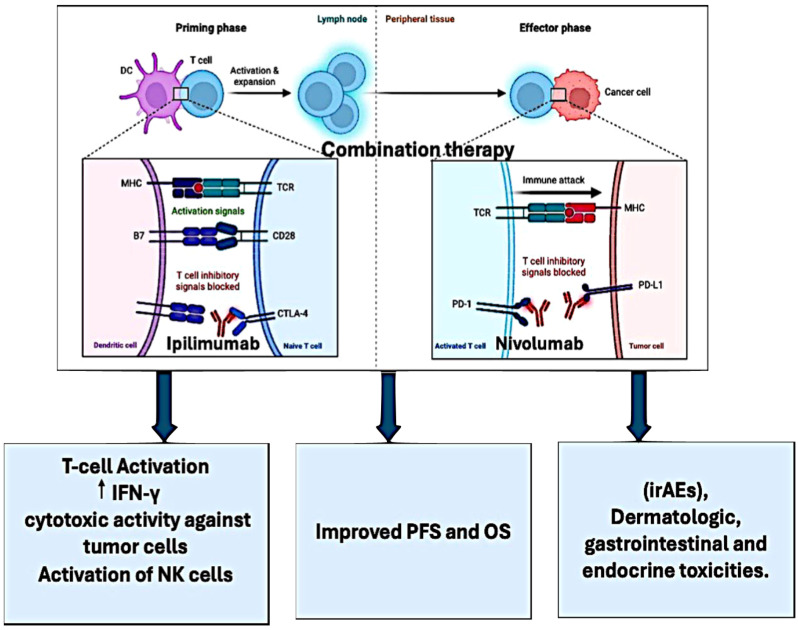
This figure illustrates the mechanisms and clinical implications of ipilimumab and nivolumab in cancer therapy. Ipilimumab inhibits CTLA-4, enhancing T cell activation, while nivolumab blocks PD-1/PD-L1 interaction, enabling activated T cells to target tumor cells. Combination therapy improves PFS and OS, enhancing T cell activation, IFN-γ cytotoxic activity, and NK cell activation, despite potential irAEs and toxicities.

**Figure 2 pharmaceutics-16-00732-f002:**
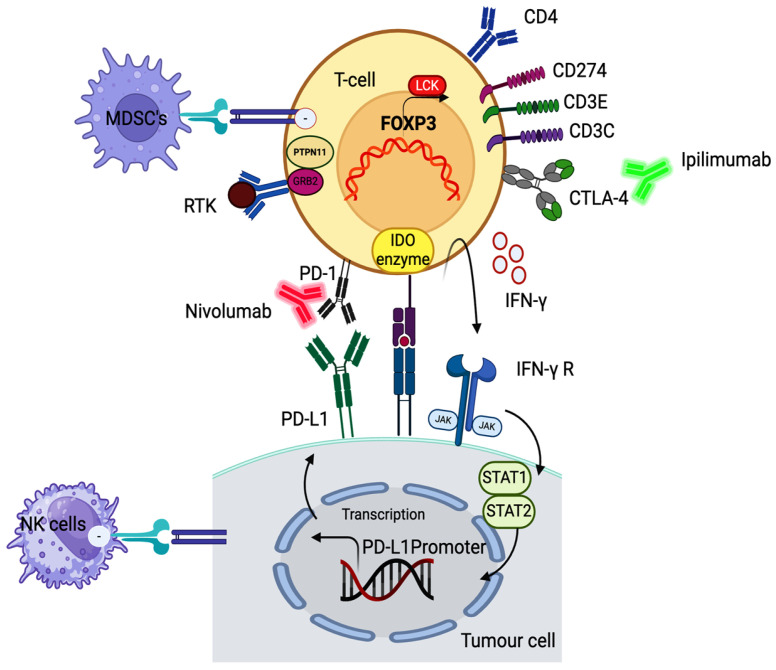
Signaling pathways inhibited by ipilimumab and nivolumab. Ipilimumab inhibits the CTLA-4 receptor on T cells, preventing its interaction with CD80/CD86 on antigen-presenting cells (APCs). Additionally, it disrupts the association between CTLA-4 and the LCK protein kinase, impacting downstream signaling events crucial for T cell activation. The interference with RTK activation domain phosphorylation and disruption of associations with CD3ζ, CD3e, and CD274 further contribute to its influence on T cell activation. Moreover, ipilimumab inhibits the association of CTLA-4 with PTPN11, a tyrosine protein phosphatase involved in key cellular processes. Nivolumab operates on multiple fronts to counteract immune evasion mechanisms in cancer. Interleukin-gamma (IFN-gamma) typically induces the expression of programmed death ligand 1 (PD-L1) on the surface of tumor cells, allowing them to evade immune surveillance. Nivolumab inhibits this immune escape strategy by blocking the PD-1 receptor on T cells, preventing its interaction with PD-L1 and PD-L2 on tumor cells. The inhibition of PD-1 signaling enhances NK cell cytotoxicity and promotes their antitumor functions. Additionally, nivolumab’s ability to boost T cell responses contributes to an environment less conducive to the suppressive effects of MDSCs. This blockade not only disrupts the communication between cancer cells and immune cells but also inhibits the activation of immune checkpoint pathways, boosting T cell activity against cancer cells. While not directly targeting the indoleamine 2,3-dioxygenase (IDO) enzyme, known for its role in immune tolerance, nivolumab indirectly impacts it by enhancing T cell responses, contributing to a comprehensive strategy to unleash the immune system against cancer.

**Figure 3 pharmaceutics-16-00732-f003:**
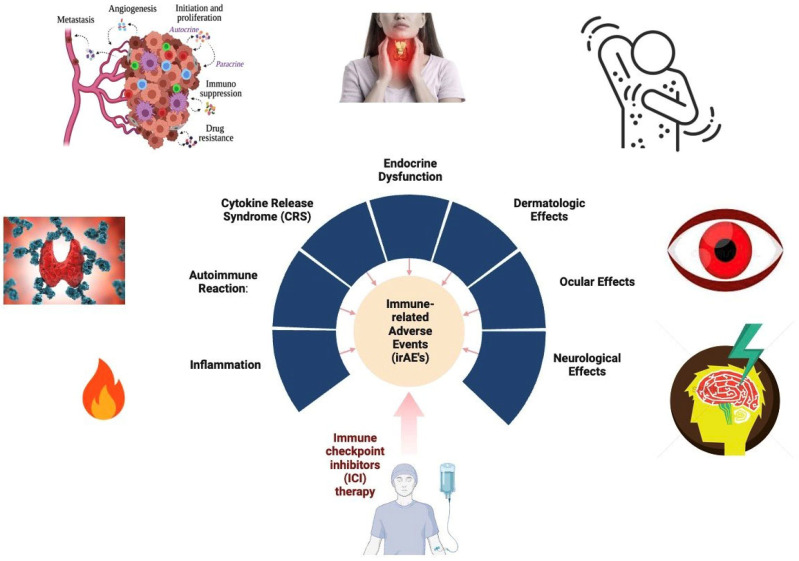
Reasons for side effects associated with nivolumab and ipilimumab combination therapy.

**Table 1 pharmaceutics-16-00732-t001:** Summary of completed clinical trials of ipilimumab plus nivolumab in non-small cell lung carcinoma [49,52,53,54,58,61,62].

Reference	Trial Phase	Treatment Arms	Primary Endpoints	Results	Clinical Trial Number
Hellmann, M. D., et al. (2019)	Phase 3	Nivolumab + Ipilimumab, Nivolumab, Chemotherapy	Overall Survival	Nivolumab + ipilimumab led to longer overall survival than chemotherapy in patients with advanced NSCLC, irrespective of PD-L1 expression.	NCT02477826
Reck, M., et al. (2021)	Phase 3	Nivolumab + Ipilimumab + 2 Cycles of Chemo, Chemotherapy (4 Cycles)	Overall Survival	Nivolumab plus ipilimumab with two cycles of chemotherapy showed durable efficacy benefits over chemotherapy alone in advanced NSCLC.	NCT03215706
Cascone, T. et al. (2021)	Phase 2	Nivolumab vs. Nivolumab + Ipilimumab	Major Pathologic Response (MPR)	Nivolumab + ipilimumab arm achieved a 38% MPR rate, with higher pathologic complete response rates and less viable tumor compared to nivolumab alone.	NCT03158129
Ready, N. E. et al. (2022)	Phase 3B	Nivolumab + Ipilimumab	Incidence of Grade 3–4 and Grade 5 Immune-Mediated Adverse Events (IMAEs)	Manageable safety and durable efficacy observed in patients with ECOG PS 0–1. Special populations, including those with ECOG PS 2 or untreated brain metastases, showed comparable safety and encouraging 3-year overall survival rates.	NCT02869789
Owonikoko, T. K., et al. (2021)	Phase 3	Nivolumab plus Ipilimumab; Nivolumab; Placebo	Overall Survival (OS)	OS not significantly prolonged with nivolumab plus ipilimumab versus placebo (HR = 0.92).	NCT02538666

**Table 3 pharmaceutics-16-00732-t003:** Summary of completed clinical trials of ipilimumab plus nivolumab in renal cell carcinoma [73,76,78,81,82].

Reference	Trial Phase	Treatment Arms	Primary Endpoints	Results	Clinical Trial Number
Motzer et al. (2019)	Phase 3	NIVO + IPI vs. SUN	OS, PFS, Objective response	NIVO + IPI demonstrated superior OS, PFS, and objective response compared to SUN in patients with previously untreated advanced renal cell carcinoma across all risk categories.	NCT02231749
Motzer et al. (2023)	Phase 3	NIVO + IPI vs. Placebo	DFS	NIVO + IPI did not improve disease-free survival versus placebo in localized renal cell carcinoma post-nephrectomy.	NCT03138512
Tykodi et al. (2022)	Phase 3b/4	NIVO + IPI	Incidence of grade ≥ 3	Nivolumab plus ipilimumab for advanced non-clear-cell RCC showed no new safety signals and encouraging antitumor activity.	NCT02982954
Rini et al. (2022)	Phase 3	NIVO + IPI vs. SUN	OS, PFS, ORR	NIVO + IPI showed superior OS, PFS, and ORR over SUN in patients with sRCC, regardless of PD-L1 expr.	NCT02231749
Grimm et al. (2023)	Phase 2	NIVO/NIVO + IPI	Objective response rate	Nivolumab induction with or without nivolumab plus ipilimumab boosts showed improved objective response rates compared to nivolumab monotherapy. Overall efficacy was inferior to upfront nivolumab plus ipilimumab.	NCT02917772

**Table 4 pharmaceutics-16-00732-t004:** Summary of completed clinical trials of ipilimumab plus nivolumab in melanoma [8,9,85,90,94,95].

Reference	Trial Phase	Treatment Arms	Primary Endpoints	Results	Clinical Trial Number
Hodi FS et al. (2018)	Phase 3	Nivolumab + Ipilimumab	Overall survival, Progression-free survival	Improved overall survival and progression-free survival with nivolumab + ipilimumab compared to ipilimumab alone.	NCT01844505
Ascierto PA et al. (2020)	Phase 2	Nivo + Ipi	Response rate by RECIST 1.1 at week 12	In total, 48% of patients had favorable antitumor effect at week 6, with 52% and 80% estimated 18-month PFS and OS, respectively.	NCT03122522
Ascierto PA et al. (2023)	Phase 2	A: Encorafenib + Binimetinib -> Ipilimumab + Nivolumab -> Nivolumab	Overall survival at 2 years	Median OS not reached in any arm; 2-year OS rates: Arm A—65%, Arm B—73%, Arm C—69%. No new safety signals emerged.	NCT02631447
Postow et al., 2021	Phase 2	Nivolumab + Ipilimumab vs. Nivolumab Alone	Response rate by RECIST 1.1 at week 12, Progression-free survival (PFS), Overall survival (OS), Safety	Best overall response rates by RECIST at week 12 or any time afterward were 48% and 58%, respectively. The 18-month progression-free survival and overall survival were estimated at 52% and 80%, respectively. Fifty-seven percent of patients had grade 3–5 treatment-related toxicity.	NCT03122522
Reijers, I. L. M., et al. (2022)	Phase 3	Neoadjuvant Ipilimumab and Nivolumab	Pathologic response rates (pRRs)	pRR was 72%, with 61% achieving major pathologic response (MPR). TLND omission was feasible for MPR.	NCT02977052
Diefenbach, C. S. et al. (2020)	Phase 1/2	Brentuximab Vedotin + Nivolumab + Ipilimumab	Safety and activity evaluation	Evaluation of brentuximab vedotin combined with nivolumab or ipilimumab, or both, in patients with relapsed or refractory Hodgkin lymphoma.	NCT01896999

**Table 5 pharmaceutics-16-00732-t005:** Summary of completed clinical trials of ipilimumab plus nivolumab across different cancer types [23,96,97,98,99,100,101,102,103,104,105,106,107,108].

Reference	Trial Phase	Treatment Arms	Primary Endpoints	Results	Clinical Trial Number
Lenz et al. (2022)	Phase 2	NIVO + low-dose IPI	Objective response rate	First-line nivolumab plus low-dose ipilimumab showed a 69% objective response rate and 84% disease control rate in MSI-H/dMMR metastatic colorectal cancer. Clinical benefit was observed regardless of baseline characteristics.	NCT02060188
Patel, S. P., et al. (2020)	Phase 2	Ipilimumab plus nivolumab	Overall response rate	ORR: 25% with 44% in high-grade neuroendocrine carcinoma; 0% in low/intermediate-grade tumors.	NCT02060188
Yau, T. et al. (2020)	Phase 1/2	Nivolumab plus ipilimumab	Safety, tolerability, objective response rate	Promising ORRs (27–32%) observed across treatment arms in advanced HCC. Arm A showed highest median overall survival (22.8 months). Manageable safety profile.	NCT01658878
Grimm, M.-O., et al. (2023)	Phase 2	Tailored immunotherapy approach with nivolumab with or without ipilimumab	Objective response rate (ORR)	Objective response rate exceeding 20% in metastatic urothelial patients.	NCT03219775
Adams, S. et al. (2021)	Phase 2	Ipilimumab + nivolumab	Objective response rate (ORR)	ORR was 18%, with 3 of 17 patients achieving objective responses (1 complete, 2 partial responses) in advanced/metastatic metaplastic breast cancer.	NCT02834013

**Table 6 pharmaceutics-16-00732-t006:** The incidence of irAEs induced by immune checkpoint inhibitors [117,118,124,125,126].

Immune-Related Adverse Event	Nivolumab (%)	Ipilimumab (%)	Combination (Nivolumab + Ipilimumab) (%)
Diarrhea	10%	23%	35%
Fatigue	15%	30%	45%
Pruritus	8%	20%	30%
Rash	12%	25%	38%
Nausea	7%	18%	28%
Pyrexia (Fever)	5%	15%	25%
Decreased appetite	6%	12%	20%
Vomiting	4%	10%	18%
Hypothyroidism	2%	8%	15%
Colitis	Less than 1%	15%	25%
Arthralgia	Less than 1%	12%	22%
Headache	Less than 1%	10%	18%
Neurological disorder	Less than 1%	8%	15%
Sweet’s syndrome	Less than 1%	-	10%
Dyspnea (Breathing difficulty)	Less than 1%	5%	12%
Liver toxicity	Less than 1%	Less than 1%	8%

**Table 7 pharmaceutics-16-00732-t007:** Summary of Side Effects, and Underlying Reasons, of Nivolumab and Ipilimumab Combination Therapy [117,118,119,120,122,123,124,125,126,127].

Side Effect	Underlying Reason
Fatigue	Activation of immune system leading to generalized tiredness
Rash	Cutaneous immune response against normal skin tissue
Diarrhea	Immune-mediated inflammation of the gastrointestinal tract
Pruritus	Activation of immune cells in the skin
Colitis	Inflammation of the colon due to immune system attack
Hepatitis	Immune-mediated inflammation of liver cells
Hypothyroidism	Autoimmune destruction of thyroid tissue
Pneumonitis	Immune-mediated inflammation of lung tissue
Nephritis	Immune-mediated inflammation of kidney tissue
Endocrinopathies	Dysfunction of endocrine glands due to immune system dysregulation
Dermatitis	Immune response leading to skin inflammation
Arthralgia	Immune-mediated joint inflammation
Myalgia	Muscle pain due to immune system activation
Neuropathy	Immune-mediated damage to peripheral nerves
Ocular toxicity	Inflammatory response affecting the eyes
Cardiotoxicity	Immune-related damage to the heart
Renal toxicity	Immune-mediated kidney damage
Hepatotoxicity	Liver damage caused by immune response
Gastrointestinal	Immune-related inflammation of the digestive tract
Cutaneous reactions	Immune response leading to skin rashes and other dermatological issues

## Data Availability

Not applicable.

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
