# Peer review of "Nivolumab and Ipilimumab Acting as Tormentors of Advanced Tumors by Unleashing Immune Cells and Associated Collateral Damage"

_pharmaceutics, 2024, doi:10.3390/pharmaceutics16060732_

Round 1

Reviewer 1 Report

Comments and Suggestions for Authors

In their review, Khan et al discuss the mechanisms, adverse effects, and various clinical trials involving nivolumab and ipilimumab across different cancers to highlight the profound impact their combination has had on cancer treatment. While this review presents a detailed insight on the ICB combinations, some concerns remain:

1. introduction: to improve the flow, the authors could consider introducing nivolumab after ipilimumab before introducing their combination.

2. Similarly to improve flow and provide better structure for easier read, authors could consider combining headings 2 and 3 under Ipilimumab, and 4 and 5 under nivolumab.

3. Please move the second paragraph under heading 2 (‘Combining nivolumab and ipilimumab as a first-line therapy for individuals at medium and low risk was found to be efficacious in a clinical investigation….) to separate heading after nivolumab – that way the agents and their mechanism of action is discussed followed by their combined effect.

 4. References are missing through out the manuscript (eg: “The combination of nivolumab + ipilimumab remained superior to tyrosine kinase inhibitors in long-term follow-up for patients with advanced renal cell carcinoma”, “The combination of nivolumab and ipilimumab consistently had a higher response rate than tyrosine kinase inhibitor” etc). Please keep the in-text citation after the relevant sentence rather than at the end of the paragraph.

 5. ‘Clinical trials’ should be remained as ‘Clinical trials of combination’. Also on what basis where the cancers discussed in section 7 selected. What about CRC, breast, Head and neck, HCC, urothelial, Hodgekin’s lymphoma? Clinical trial numbers or their published reports should be included as references throughout this section.

 6. Authors could consider providing clinical trial details (including trial phase, cancer type, results, treatment arms, endpoints, year etc.) as a table for greater clarity either.

 7. for Clinical trial section – consider providing PFS/OS of combination vs monotherapy where available to highlight impact of combination treatment.

 8. As MDSCs are relevant to other cancers as well – the authors could consider providing a more pan-cancer overview of MDSCs rather than limiting their description to melanoma alone.

 9. The discussion of auto-abs in correlation to the two ICI discussed is not clear – it may be better to remove this paragraph and focus instead Neurological irAEs, adverse effects on CNS and PNS (eg: PMID: 31653079).

 10. Authors could consider including a table comparing incidence of common irAEs in each ICI and their combination.

11. Authors should include a section discussing resistance to ICI considering that a significant proportion of patients fail to respond to ICIs

12.The authors could consider adding a table demonstrating potential trials/studies showing benefits of nivolumab + ipilimumab combination with other agents. 

 Minor

1. This sentence is unclear ‘2018 saw the standard treatment for metastatic renal cell carcinoma become tyrosine kinase inhibitors, which took the place of the prior first-line therapy’ – do the authors mean ipilimumab was the prior first-line therapy before being replaced by TKI?

2. Why is sweet syndrome mentioned but not discussed in adverse effect sections?

3. Please check manuscript for typos (apilimumab, syn-drome)

4. All abbreviations should be mentioned in full when first used (Immune checkpoint inhibitors (ICIs) – has been mentioned as such in multiple places)

Comments on the Quality of English Language

Overall the manuscript reads well. Minor proofreading is required to improve clarity of few sentences and omit typos

Author Response

Responses to Our Academic Editor and Reviewer(s)

Reviewer 1 comments: In their review, Khan et al discuss the mechanisms, adverse effects, and various clinical trials involving nivolumab and ipilimumab across different cancers to highlight the profound impact their combination has had on cancer treatment. While this review presents a detailed insight on the ICB combinations, some concerns remain:

  1. introduction: to improve the flow, the authors could consider introducing nivolumab after ipilimumab before introducing their combination.

Authors: Certainly, I have made the suggested revision in the introduction of the manuscript. By introducing nivolumab after ipilimumab before discussing their combination.

  1. Similarly to improve flow and provide better structure for easier read, authors could consider combining headings 2 and 3 under Ipilimumab, and 4 and 5 under nivolumab.

Authors: I have combined headings 2 and 3 under Ipilimumab, and 4 and 5 under Nivolumab to improve the flow and provide better structure for easier reading, as suggested.

  1. Please move the second paragraph under heading 2 (‘Combining nivolumab and ipilimumab as a first-line therapy for individuals at medium and low risk was found to be efficacious in a clinical investigation….) to separate heading after nivolumab – that way the agents and their mechanism of action is discussed followed by their combined effect.

Authors: I have moved the second paragraph under heading 2, as requested. Now, the agents and their mechanism of action are discussed first, followed by their combined effect.

Top of Form

Bottom of Form

  1. References are missing through out the manuscript (eg: “The combination of nivolumab + ipilimumab remained superior to tyrosine kinase inhibitors in long-term follow-up for patients with advanced renal cell carcinoma”, “The combination of nivolumab and ipilimumab consistently had a higher response rate than tyrosine kinase inhibitor” etc). Please keep the in-text citation after the relevant sentence rather than at the end of the paragraph.

Authors: I have added in-text references throughout the manuscript, as suggested.

  1. ‘Clinical trials’ should be remained as ‘Clinical trials of combination’. Also on what basis where the cancers discussed in section 7 selected. What about CRC, breast, Head and neck, HCC, urothelial, Hodgekin’s lymphoma? Clinical trial numbers or their published reports should be included as references throughout this section.

Authors: I have made the changes accordingly. The heading "Clinical trials" has been adjusted to "Clinical trials of combination," and additional information about CRC, breast, Head and neck, HCC, urothelial, and Hodgkin’s lymphoma has been included in subsection 7.5. Moreover, clinical trial numbers have been added to the tables for reference throughout the sections.

  1. Authors could consider providing clinical trial details (including trial phase, cancer type, results, treatment arms, endpoints, year etc.) as a table for greater clarity either.

Authors: I have already provided tables for each subsection in section 7, including all the details you mentioned, such as trial phase, cancer type, results, treatment arms, endpoints, and year. This should enhance the clarity and organization of the information presented in the manuscript.

  1. for Clinical trial section – consider providing PFS/OS of combination vs monotherapy where available to highlight impact of combination treatment.

Authors: I have incorporated PFS/OS data comparing combination therapy with monotherapy in several clinical trials within the Clinical Trial section

  1. As MDSCs are relevant to other cancers as well – the authors could consider providing a more pan-cancer overview of MDSCs rather than limiting their description to melanoma alone.

Authors: I have addressed this concern by expanding the discussion on myeloid-derived suppressor cells (MDSCs) beyond melanoma alone. While research on MDSCs in the context of nivolumab and ipilimumab combination therapy in pan-cancer settings is currently limited, I have included additional text discussing their relevance in lung carcinoma and preclinical studies

  1. The discussion of auto-abs in correlation to the two ICI discussed is not clear – it may be better to remove this paragraph and focus instead Neurological irAEs, adverse effects on CNS and PNS (eg: PMID: 31653079).

Authors: I have addressed this concern by removing the paragraph discussing auto-antibodies in correlation to the two immune checkpoint inhibitors (ICIs) discussed. Instead, I have focused on adverse events, particularly neurological immune-related adverse events (irAEs), and adverse effects on the central nervous system (CNS) and peripheral nervous system (PNS) in sections 8 of the manuscript

  • Authors could consider including a table comparing incidence of common irAEs in each ICI and their combination.

 Authors: I have addressed this suggestion by creating a table comparing the incidence of common immune-related adverse events (irAEs) in each immune checkpoint inhibitor (ICI) alone and their combination therapy of ipilimumab and nivolumab.

  • Authors should include a section discussing resistance to ICI considering that a significant proportion of patients fail to respond to ICIs

Authors: I have addressed this suggestion by adding a section discussing resistance to immune checkpoint inhibitors (ICI) in the manuscript, specifically focusing on resistance to combination therapy in the 7.6 subsection.

12.The authors could consider adding a table demonstrating potential trials/studies showing benefits of nivolumab + ipilimumab combination with other agents.

Authors: I have addressed this suggestion by incorporating tables in the manuscript, specifically Tables 1-5, which discuss various studies demonstrating the benefits of nivolumab + ipilimumab combination with other agents.

 Minor

  1. This sentence is unclear ‘2018 saw the standard treatment for metastatic renal cell carcinoma become tyrosine kinase inhibitors, which took the place of the prior first-line therapy’ – do the authors mean ipilimumab was the prior first-line therapy before being replaced by TKI?

Authors: I have corrected it in the sentence to make it more clearer.

  1. Why is sweet syndrome mentioned but not discussed in adverse effect sections?

Authors: I have included Sweet syndrome in section 8 where adverse effects are discussed.

  1. Please check manuscript for typos (apilimumab, syn-drome)

Authors: I have corrected it.

  1. All abbreviations should be mentioned in full when first used (Immune checkpoint inhibitors (ICIs) – has been mentioned as such in multiple places)

Authors: I have corrected it by expanding the abbreviation "ICIs" to "Immune checkpoint inhibitors" upon first use.

Reviewer 2 Report

Comments and Suggestions for Authors
The authors of the manuscript "Nivolumab and Ipilimumab acting as tormentor of advanced tumors by unleashing immune cells and associated collateral damage" embarked on writing about a challenging but highly relevant topic.
My main concern is that those who know little or no immunology have no chance of understanding the science behind the therapy and hence the manuscript. Unfortunately, the current structure of this otherwise very well written manuscript is lacking figures and tables, which does not help the reader.

To make sure that it is clear for the authors what I would find essential are as follows:

The introduction is good, it focuses the reader's interest, but a simple figure would help.

Being an immunologist who also teaches pharmacy students about immune checkpoint therapy, I would strongly suggest a bit of restructuring the rest of the manuscript: 

1) It would help to start with the role of both PD1/PDL1 and CTLA4/B7. What is their basic role in keeping immune activity at bay.

2) Although PD1 and CTLA4 inhibit T cell activation, clarifying where the molecules are located would be helpful and demonstrating it in a figure. That would help to emphasize the significance of the discovery of ICT.

3) Mode of action with less busy figure would also be necessary, potentially with an added table

4) Reasons for side effects should be summarized in a table and explained in a figure.

The manuscript refers to recent articles and clinical trials which makes the review a valuable work.

Comments on the Quality of English Language

Quality of English is good. 

Author Response

Reviewer 2 comment: The authors of the manuscript "Nivolumab and Ipilimumab acting as tormentor of advanced tumors by unleashing immune cells and associated collateral damage" embarked on writing about a challenging but highly relevant topic.
My main concern is that those who know little or no immunology have no chance of understanding the science behind the therapy and hence the manuscript. Unfortunately, the current structure of this otherwise very well written manuscript is lacking figures and tables, which does not help the reader.

To make sure that it is clear for the authors what I would find essential are as follows:

The introduction is good, it focuses the reader's interest, but a simple figure would help.

Being an immunologist who also teaches pharmacy students about immune checkpoint therapy, I would strongly suggest a bit of restructuring the rest of the manuscript: 

1) It would help to start with the role of both PD1/PDL1 and CTLA4/B7. What is their basic role in keeping immune activity at bay.

Authors: Certainly, I've included a discussion on the roles of PD1/PDL1 and CTLA4/B7 in the introduction section as suggested

2) Although PD1 and CTLA4 inhibit T cell activation, clarifying where the molecules are located would be helpful and demonstrating it in a figure. That would help to emphasize the significance of the discovery of ICT.

Figure 1: This figure illustrates the mechanisms and clinical implications of ipilimumab and nivolumab in cancer therapy. Ipilimumab inhibits CTLA-4, enhancing T cell activation, while nivolumab blocks PD-1/PD-L1 interaction, enabling activated T cells to target tumor cells. Combination therapy improves PFS and OS, enhancing T cell activation, IFN-γ cytotoxic activity, and NK cell activation, despite potential irAEs and toxicities.

Authors: I've incorporated a figure that clarifies the locations of PD1 and CTLA4 in the manuscript.

3) Mode of action with less busy figure would also be necessary, potentially with an added table

Authors : Figure has been added.

4) Reasons for side effects should be summarized in a table and explained in a figure.

Authors: I have implemented both recommendations. The reasons for side effects have been summarized in a table for easy reference, and an accompanying figure has been provided to visually explain these reasons.

The manuscript refers to recent articles and clinical trials which makes the review a valuable work.

Round 2

Reviewer 1 Report

Comments and Suggestions for Authors

Authors have adequately addressed the points raised by the reviewer. Minor comment: for clarity - the authors could add ‘cancer’ as a column heading and state the cancer there instead of under ‘results’ in table 5.